# Dynamic Calibration Method of Sensor Drift Fault in HVAC System Based on Bayesian Inference

**DOI:** 10.3390/s22145348

**Published:** 2022-07-18

**Authors:** Guannan Li, Haonan Hu, Jiajia Gao, Xi Fang

**Affiliations:** 1School of Urban Construction, Wuhan University of Science and Technology, Wuhan 430065, China; leegna@163.com (G.L.); huhaonan2046@163.com (H.H.); 2College of Civil Engineering, Hunan University, Changsha 410082, China; xfang@hnu.edu.cn

**Keywords:** sensor drift fault, Bayesian inference (BI), field dynamic calibration, fault detection, HVAC system

## Abstract

Sensor drift fault calibration is essential to maintain the operation of heating, ventilation and air conditioning systems (HVAC) in buildings. Bayesian inference (BI) is becoming more and more popular as a commonly used sensor fault calibration method. However, this method focused mainly on sensor bias fault, and it could be difficult to calibrate drift fault that changes with time. Therefore, a dynamic calibration method for sensor drift fault of HVAC systems based on BI is developed. Taking the drift fault calibration of the chilled water supply temperature sensor of the chiller as an example, the performance of the proposed dynamic calibration method is evaluated. Results show that the combination of the Exponentially Weighted Moving-Average (EWMA) method with high detection accuracy and the proposed BI dynamic calibration method can effectively improve the calibration accuracy of drift fault, and the Mean Absolute Percentage Error (MAPE) value between the calibrated and normal data is less than 5%.

## 1. Introduction

Building energy consumption is huge, which is close to 40% of the global energy consumption. In the next 20 years, this proportion will increase by more than 1.5% every year [1]. As the largest energy consumer in the construction field, the energy consumption of HVAC systems accounts for more than 40% of total building energy consumption [2]. Therefore, improving the energy efficiency of HVAC systems is one of the key ways to reduce building energy consumption. At present, energy-saving technologies such as system operation diagnosis, building automatic control and building energy management [3] have been widely used to improve the energy efficiency of HVAC systems. The application of these technologies is based on the measurement data of various sensors in the system [4]. However, with the passage of time, the sensors used to detect and record system operation data may inevitably suffer various types of faults in their whole life cycles. Sensor faults not only affect the energy-saving technology, increase the energy consumption of HVAC systems [5,6], but also reduce indoor thermal comfort directly [7,8]. Among them, as one of the common sensor fault types, drift fault is very harmful to the system because its fault offset changes with time. Hence, the sensor drift fault calibration of HVAC system sensors is very important.

In practical application, the field calibration of sensors after a fault usually includes two steps: fault detection and fault calibration. Sensor fault detection methods are mainly divided into three categories: quantitative model-based methods, qualitative model-based methods and data-driven methods. In recent years, due to the accelerated upgrading of computer software and hardware and the rapid development of data storage and transmission technology, data-driven fault detection methods have attracted more and more attention [9]. The advantages of this data-driven method are: firstly, it is very suitable for the problem of insufficient interpretation of the physical model [10]. Secondly, compared with the complex model-based method, the data-driven method does not involve the physical model of the system and is easy to build. Thirdly, the statistical method constructed by simple algorithms can significantly reduce the amount of calculation and shorten the calculation time. Chang et al. [11] processed the original data training set by using the method of the Laida criterion to detect the fault data in the original data. This method is one of the common methods to detect faults [12]. Chang et al. [13] used the Box-plot method for data screening in the prediction model. This method can be used not only to detect the abnormal value of sample data subject to a normal distribution, but also detect the abnormal values of the sample data that do not obey the normal distribution, which has a wide range of applications [14]. Zhao [15] proposed an exponentially weighted moving average control chart (EWMA) method to detect sensor faults in centrifugal coolers. This method considers both current samples and previous samples and can quickly detect small fluctuations in the sensor array. As the drift fault offset gradually increases from zero, the actual fault detection method may not successfully detect some fault data for some time. Considering the aforementioned studies, this study attempts to make a comparison among the three detection methods and finds out the method with higher detection accuracies.

For fault calibrations, the most used method is the on-site calibration method [16,17]. This method adopts the intensive deployment of sensors and uses the relevant formulas to calculate according to the readings and known conditions of the reference sensor. The sensor calibration is realized by eliminating the unknown parameters in the calibration environment. This type of sensor calibration methods is suitable for the situation where the sensors are interrelated, or there is sensor redundancy. However, for the actual HVAC system, considering the installation time and cost of sensors, the impact of interrupting the system during calibration and the difficulty of replacing the sensors inside the wall, the traditional on-site offline calibration technology is difficult to apply [18]. In recent years, the Bayesian inference (BI) method has been widely used to calibrate sensor faults in HVAC systems. This method can realize online fault detection and calibration, effectively removing the adverse effects caused by sensor faults on the basis of uninterrupted system operation and effectively solving the problems existing in on-site calibration. Mokhtar et al. [19] calibrated the wind speed sensor in a thermal power plant by the BI method. Yoon [20] proposed a virtual field sensor calibration method for building energy systems. The BI method was used to derive the optimal solution in the calibration process. Wang et al. [21] used the BI method to calibrate the sensors in a photovoltaic heat pump system. Choi and Yoon [22] combined the virtual sensor with the BI method to calibrate the bias fault of the sensor in the air handling unit (AHU). Zhao et al. [23] used the method of variable BI expectation maximization to calibrate the four sensor faults in AHU. Liu et al. [24] developed the sensor fault diagnosis and self-calibration method based on BI and virtual sensing coupling, which effectively identified the operating state of the system and accurately identified the fault location so as to achieve self-calibration. Kim et al. [25] eliminated the sensor error in the variable air volume (VAV) system by the BI-based sensor calibration method.

The existing fault calibration methods mentioned above are mostly aimed at the sensor bias faults. However, the sensor drift fault is more complex than the bias fault. The initial fault bias amplitude is usually small, but as time goes by, the fault bias amplitude increase gradually, and the harm to the system increase enventually. At present, many researchers have studied drift fault calibration. Jana et al. [26] used a convolutional automatic encoder (CAE) to calibrate the drift of a single sensor, and the error after calibration was less than 1%. Wang and Yang [27] used the residual signal iterative estimation method to calibrate the drift of the temperature sensor. Yue et al. [28] used the improved principal component analysis (PCA) model to calibrate the sensor drift fault of a nuclear power plant, and the error was less than 5%. However, these studies did not cover the direct calibration of drift fault through BI. The reason could be that the traditional and orginal BI method could be a time-independent sensor fault calibration method. The calibrated value is a time-independent fixed value, which may be very difficult to solve the problem that the sensor offset changes with time. So This means the original BI may be not suitable for directly application on sensor drift fault calibration for HVAC systems.

Based on the aforementioned literature review, this study presents a dynamic calibration method of sensor drift fault in an HVAC system based on Bayesian inference. In this method, the time characteristic variable is introduced to deal with the dynamic characteristics of the drift, and then the dynamic calibration of the BI based sensor drift fault is realized using both simulation and practical cases, combined with the data-driven fault detection algorithm. Taking the drift fault calibration of the chilled water supply temperature sensor of the chiller as an example, the performance of the proposed dynamic calibration method is evaluated by introducing different drift fault amplitudes, and the influence of time length on the calibration accuracy is further studied. The main contributions of this paper could be: (1) introducing time characteristic variables into Bayesian inference to deal with the dynamic characteristics of drift so as to realize the dynamic calibration of sensor drift fault; (2) combining the proposed dynamic calibration method with EWMA method with high detection accuracy, the time from fault to fault calibration is effectively shortened, and the calibration accuracy of sensor drift fault is improved; (3) the relationship between calibration time length and calibration accuracy is given.

The paper is organized as follows: Section 2 introduces three detection algorithms and the basic principle of the dynamic calibration method based on Bayesian inference. Section 3 introduces the research framework of this paper. Section 4 gives a description of the system used for the case study. Section 5 presents the calibration effect of the proposed dynamic calibration method in case application. Section 6 compares the dynamic calibration method based on Bayesian inference with the principal component analysis method. Section 6 compares this method with the least square method. Section 7 is the conclusion.

## 2. Dynamic Calibration Process of Sensor Drift Fault in HVAC System Based on Bayesian Inference

### 2.1. Principles of Three Sensor Fault Detection Methods

This study adopts the Laida criterion, Box-plot and EMWA control chart detection algorithm. These are implemented on the python platform.

#### 2.1.1. Laida Criterion

The Laida criterion [12] takes the standard deviation δ of three times the normal value as the selection standard. The detection algorithm has the advantages of a simple model, small computation and good fault detection performance. Equation (1) for standard deviation δ is as follows, it is the parameter calculated after a large number of repeated observations and its calculation Equation (1) is as follows:(1)$$\delta =\sqrt{\frac{\sum_{i=1}^{n}\left(Y_{n}-\bar{Y}\right)}{n-1}}$$

In Equation (1), n is the number of observed values in the HVAC system, $Y_{n}$ represents normal data and $\bar{Y}$ is the arithmetic mean of normal data.

The calculation of its threshold is shown in Equation (2):(2)$$UCL_{Laida}\backslash LCL_{Laida}=\bar{Y}\pm 3\delta$$
where $UCL_{Laida}$ and $LCL_{Laida}$ are the upper and lower limits of the Laida criterion, respectively. If the HVAC system test data is greater than $UCL_{Laida}$ or less than $LCL_{Laida}$, the sensor has failed.

#### 2.1.2. Box-Plot

Box-plot [13] is an effective graph for processing data. Its algorithm is efficient. This method uses all kinds of quantiles (upper quartile, median and lower quartile) of the sample to establish the quantile distance and then establish the corresponding threshold. The specific formulas are shown in Equations (3) and (4).
(3)$$UCL_{Box-plot}=Y_{3}+1.5Y_{3}-Y_{1}$$
(4)$$LCL_{Box-plot}=Y_{1}-1.5\left(Y_{3}-Y_{1}\right)$$
where, $Y_{3}$ and $Y_{1}$ are the upper quartile and lower quartile of normal data of the HVAC and R systems in ascending order, respectively. If the test data of the HVAC system is greater than $UCL_{\mathrm{Box}-\mathrm{plot}}$ or less than $LCL_{\mathrm{Box}-\mathrm{plot}}$, it indicates that the sensor has failed.

#### 2.1.3. EWMA Control Diagram

EWMA control chart [15] is widely used to monitor the operation process of industrial equipment because of its simple detection model, easy construction and small amount of calculations. For the system operation process data, the EWMA control chart algorithm allocates the corresponding weight factors on the basis of fully considering the importance of data at the current time and the previous time and constructs the corresponding upper or lower limit of control limit to effectively detect abnormal small fluctuations. Its calculated value represents the time weight mean of all data in the monitoring process. EWMA calculated values are usually defined as:(5)$$Z_{m}=\left(1-\lambda \right)Z_{m-1}+\lambda y_{m}$$

In Equation (5) $Z_{m}$ is the m-th EWMA calculated value, $Z_{m-1}$ is the m-1st EWMA calculated value, $y_{n}$ is the nth data, and N represents the number of iterations, *λ* Is the weight factor (0 ≤ *λ* ≤ 1). Usually, a smaller *λ* will lead to smaller displacement and a faster detection process [29]. The best *λ* is usually between 0.2 and 0.3 [30]; in this study, *λ* = 0.2, and the number of iterations is 6305. The control limit definition of the EWMA control chart is shown in Equation (6):(6)$$UCL\backslash LCL_{EWMA}=y_{0}\pm L_{0}\delta \sqrt{\frac{\lambda }{\left(2-\lambda \right)}\left[1-{\left(1-\lambda \right)}^{2n}\right]}$$

In Equation (6), factor *L* represents the width of the control limit, *L* = 3. δ is the standard deviation of data, and n represents the number of data. When the observed value y is greater than $UCL_{EWMA}$ or less than $LCL_{EWMA}$, it indicates that the numeric is a fault numeric.

### 2.2. Dynamic Calibration Method Based on Bayesian Inference

Bayesian inference [31,32,33] is a statistical inference method that updates the probability of specific assumptions according to new information, which is widely used in many fields. In the building system, by comparing the difference between the data of the sensor under normal conditions and the data under fault conditions, it obtains the required calibration results through a series of formula derivations.

After the sensor has a bias fault, the offset between the normal value and the fault numeric is a fixed value, which can be obtained by the Bayesian inference method. However, after the sensor has drift fault, the offset between the normal value and the fault numeric will change with time, while the traditional Bayesian method cannot get the dynamic offset and cannot solve the drift fault. Therefore, this study introduces a time characteristic based on Bayesian inference and proposes a dynamic calibration method of drift fault based on Bayesian inference. The details are as follows:

The error is between normal value and fault numeric of sensor x = ($x_{1}$, $x_{2}$..., $x_{i}$), where $x_{i}$ represents the offset at time i, and π (x) represents the prior function of error x; a priori is the first guess of x probability. The system model $g\;\left(x\right)=(x_{i},\;\bar{Y})$ is constructed, and the likelihood Equation (7) in the posterior distribution function is obtained. The normalized function $P\left(Y_{b}\right)$ of the fault numeric $y_{m}$ in Equation (8) is sampled by the Markov chain Monte Carlo (MCMC) [34] algorithm so as to obtain the posterior distribution Equation (10) and the maximum posterior distribution value $x_{m}$. The time $t_{m}$ at which $x_{m}$ is located is obtained by Equation (10). The drift slope *k* is obtained by Equation (11), and then the drift fault calibration value is obtained by Equation (12):(7)$$p\left(Y_{b}|x\right)=\frac{1}{\delta \sqrt{2\pi }}\exp\left[-\frac{1}{2\delta^{2}}{\left\{y_{m}-gx\right\}}^{2}\right]$$
(8)$$P\left(Y_{b}\right)=\int P\left(Y_{b}|x\right)\pi \left(x\right)dx$$
(9)$$P\left(x|Y_{b}\right)=\frac{P\left(Y_{b}|x\right)\times \pi \left(x\right)}{P\left(Y_{b}\right)}$$
(10)$$t_{m}=F\left(x_{m}\right)$$
(11)$$k=\frac{x_{m}}{t_{m}-t_{0}}$$
where, $t_{0}$ is the time when the fault is detected.
(12)$$T_{i0}=T_{i1}-k*\left(t_{i}-t_{0}\right)$$
where $T_{i0}\;$is the calibration value at time i and $T_{i1}$ is the fault numeric at time i.

### 2.3. Lag Time and Calibration Time

The trend of drift fault is that zero increases gradually. Any actual fault detection method cannot be detected part of the time, which leads to a lag between fault detection and fault occurrence. The calibration fault needs fault data, so the calibration process will also lag behind the fault detection. Figure 1 shows the drift fault results of 0.12 °C/h, 0.18 °C/h and 0.24 °C/h under the EWMA control chart. In Figure 1, $\tau_{1}$ is the time point when the fault occurs, at which time the data begins to drift. $\tau_{2}$ is the time point when the fault of 0.24 °C/h is detected, and the period between $\tau_{1}$ and $\tau_{2}$ is the time when the fault occurs but is not detected, which is called the lag time $t_{1}$. $\tau_{3}$ is the time to start calibration. Fault calibration occurs by modeling the fault data of $\tau_{2}$ and $\tau_{3}$, which is called calibration time $t_{2}$.

## 3. Research Framework

Figure 2 shows the main framework of this study. The research framework of this study is divided into four steps: data preparation, fault detection, fault calibration and discussion of influencing factors.

Data preparation

The chilled water supply temperature sensor in the chiller is selected as the target sensor under both simulation and actual conditions. The detection algorithms are the Laida criterion, Box-plot chart and the EWMA control chart, and some or all of the data in May, June and July are selected for modeling.

Fault detection

The time point of fault is detected by three detection methods under different modeling data, and the detection algorithm with the shortest lag time and modeling data are selected to detect the fault.

Fault calibration

From a practical point of view, the mode of detection before calibration is adopted. After detection by the detection algorithm obtained in the previous section, the fault is calibrated by the dynamic calibration method to obtain the calibration value. The calibration slope and the MAPE value between the normal value and calibration value are calculated.

Influence factor

The factors influencing the calibration accuracy are discussed as follows: (1) different actual working conditions; (2) different calibration times; (3) different fault amplitudes; (4) different sampling intervals.

The calculation formula of MAPE is shown in Equations (13) and (14):(13)$$Y_{cal}=Y_{0}+kt_{2}$$
(14)$$MAPE=\frac{100\%}{n}\overset{n}{\underset{i=1}{\sum }}\left|\frac{Y_{cali}-Y_{n}}{Y_{n}}\right|$$
where, $Y_{0}$ is fault data, $Y_{cal}$ is calibration data and $t_{2}$ is calibration time.

## 4. Case Description

### 4.1. Case 1 Simulated Chiller System

The target building is a typical small office building. Figure 3 is the stereoscopic view of the target building. The single-story building has five hot areas, one core hot area in the middle and four hot areas around. The whole building has a total length of 30.5 m, a total width of 15.2 m, a total height of 3.0 m and a building area of 463.6$\;m^{2}$.

The cold source of the building’s central air conditioning is a chiller. Figure 4 shows the system diagram of the chiller. As shown in the figure, the target sensor is the chilled water supply temperature sensor. In order to better meet the actual working conditions, the working time of the simulation system is set as 7:00–18:00 of the working day, and the meteorological parameters are taken from Wuhan city. We selected part of the data of May, June and July as the normal data. The data on August 1 was taken as the test data. The fault occurred at 12:00 noon on August 1. The drift slopes were 0.12 °C/h (1.5% of the average value of the sensor under normal working conditions), 0.18 °C/h (2.25%) and 0.24 °C/h (3%), respectively. The sampling interval was once every 5 min.

### 4.2. Case 2 Actual Chiller System

The data of case 2 is the data of an actual chiller system collected in an industrial plant along the east coast of China from May to August 2018. Some data from May, June and July 2018 were selected as normal data. The data on 6 August 2018 was used as test data. The target sensor was also a chilled water supply sensor. The working hours of the system were 9:00–18:00. The fault occurred at 12:00 noon on 1 August 2018. The drift slopes were 0.12 °C/h, 0.18 °C/h and 0.24 °C/h, respectively. Data were recorded every 5 min.

## 5. Results and Discussion

### 5.1. Case 1

#### 5.1.1. Discussion on Detection Methods

Data-driven detection algorithm: EWMA control chart, Laida criterion method and Box-plot method are widely used in fault detection of HVAC systems. Therefore, these three methods were used to detect sensor data in this study in order to reduce the influence of time lag in modeling and detection. Figure 5 shows the thresholds obtained from different modeling data and different detection algorithms. The minimum threshold was obtained by the EWMA control chart algorithm through one-month modeling, and the maximum threshold was obtained by the Laida criterion algorithm through three-month modeling.

#### 5.1.2. Fault Calibration

According to the previous section, we found that the EWMA control chart method based on one-month modeling had the smallest threshold, so the fault detection time was the shortest compared with other methods. In order to improve the timeliness and accuracy of calibration, the detection method of the modeling data is combined with the dynamic calibration method based on Bayesian inference, and the EWMA control chart is used for calibration after detection. Figure 6 shows the calibration results under three drift slopes. Table 1 shows the drift slope calculated by the dynamic calibration method and the absolute percentage error (MAPE) between the calibrated data and the average. As can be seen from Figure 6, the calibrated data gradually approaches the normal value with increasing time. Finally, it remains the same as the normal data. The larger the drift slope, the smaller the lag time and the shorter the calibration time. Table 1 shows that the calculated slopes under the three drift slopes are about 0.6 °C/h, and the MAPE value increases with the increase of the drift slope.

#### 5.1.3. Relationship between Calibration Accuracy and Calibration Time

Figure 7 shows the changes in k value and MAPE value under three drift faults, and the black dotted line represents the set slope. It can be seen that the three calibration slopes and MAPE values tend to be stable with the increase in calibration time. The calibration slope and MAPE value fluctuate greatly when the calibration time is short. Furthermore, the change curves of the three drift slopes are not consistent. The calibration slope at 0.12 °C/h decreases gradually with the increasing calibration time. The change curve at 0.18 °C/h rises first and then decreases, while it rises gradually at 0.24 °C/h. All three calibration slopes are above the set slope. The trend of the MAPE curve and drift slope curve under the three fault amplitudes is opposite.

#### 5.1.4. Discussion of Sensor Sampling Interval

This section discusses the effect of sensor data sampling interval on calibration accuracy. Under the drift fault of 0.24 °C/h, the sampling interval of the sensor is 5 min, 10 min and 15 min. Figure 8 shows the changes in k value and MAPE value at three different time intervals. With the increase in calibration time, the calibration slope and MAPE value under the three time intervals tended to be the same. Under the same calibration time, the fluctuation of the k and MAPE values with a five-minute sampling interval is the smallest, followed by the sampling interval of ten minutes. The most volatile is when the sampling interval is 15 min. The shorter the calibration time, the greater the impact of sampling interval. The longer the calibration time, the smaller the impact of sampling interval on calibration.

### 5.2. Case 2

#### 5.2.1. Discussion on Detection Methods

Figure 9 shows the detection results under three drift slopes. It can be seen that the detection threshold of the EWMA control chart method is smaller than that of the other two methods. At the same time, under the same fault condition, EWMA is the first to detect the fault, with the shortest lag time, followed by the Laida criterion method, and finally the Box-plot method. Under the same detection algorithm, the lag time decreases with the increase of drift slope.

#### 5.2.2. Influence of Modeling Data on Test Results

According to the discussion in Section 5.2.1, compared with the other two methods, the EWMA control chart has the minimum threshold and the shortest lag time. Therefore, the EWMA control chart algorithm is selected for fault detection. Due to the modeling characteristics, this section selects the amount of modeling data for the EMWA control chart. The data volume of one month in July, two months in June and July and three months in May, June and July are used for modeling. Three different temperature thresholds were obtained. As shown in Figure 10, the detection results of three different modeling data quantities are shown. We found that the threshold obtained by modeling with one month of data is the smallest, followed by the threshold obtained by modeling with two months of data and the threshold obtained by modeling with three months is the largest. The first fault detected is the threshold value modeled from one month’s data.

#### 5.2.3. Fault Calibration

According to Section 5.2.1 and Section 5.2.2, the EWMA control chart method based on one month of modeling is combined with the dynamic calibration method based on Bayesian inference. Figure 11 shows the calibration results under three drift slopes, in which (a) is 0.12 °C/h drift fault, (b) is 0.18 °C/h drift fault and (c) is 0.24 °C/h drift fault. Table 2 shows the drift slope after calibration and the MAPE values of calibrated data and normal data. Figure 11 shows that the calibrated data are distributed near the normal data. It shows that this method is effective in calibrating drift fault. It is found in Table 2 that as the drift slope increases, the difference between the calculated slope and the set slope becomes smaller. At the same time, it can be seen that the MAPE value between normal data and calibration data is less than 5%.

#### 5.2.4. Relationship between Calibration Accuracy and Calibration Time

In order to consider whether the accuracy of calibration is related to the time required for calibration, the curve of drift slope k and MAPE changing with calibration time is made. Figure 12 shows the changes in k value and MAPE with calibration time under three drift slopes. As can be seen from Figure 12:Under the drift slope of 0.12 °C/h, with the increase in calibration time, the k value and MAPE showed a trend of gradually increasing at first and then tending to be stable. The slope is stable at about 0.14 °C/h, and the MAPE value is stable at about 2%.The k value of 0.18 °C/h increases with time, and the value gradually drops to the setting slope. The MAPE value gradually rises to around 4%.The k value of 0.24 °C/h rises first, then decreases, and finally tends to be near the set slope. MAPE first decreased, then increased, and finally stabilized at around 4.5%.

It can be seen from the figure that when the calibration time is short, the k value and MAPE value obtained fluctuate greatly, and the value is unstable. If high calibration accuracy is required, the calibration will take a long time.

At the same time, in order to explore the change in the calibration accuracy of the same sensor with the calibration data under different working conditions, the same type of sensor drift fault of a metro chiller is selected for verification, and the sampling interval is once every 5 min.

Figure 13 shows the changes in k value and MAPE value under a subway. It can be found that under the drift slopes of 0.12 °C/h and 0.18 °C/h, the slope after cooling tower data calibration is closer to the set slope than that after subway data calibration in the same calibration time. Under the drift slope of 0.24 °C/h, the slope after Metro data calibration is closer to the set slope. At the same time, the calibration data of Metro is smaller than the calibration data MAPE of the cooling tower in the three cases, indicating that after calibration, the data of Metro is closer to the normal data than the data of the cooling tower.

#### 5.2.5. Discussion of Sensor Sampling Interval

This section discusses the influence of sensor data sampling interval on calibration accuracy. Under the drift fault of 0.24 °C/h, the data shall be recorded every 5 min, 10 min and 15 min. Figure 14 shows the changes in k value and MAPE value at three different time intervals. It can be found that under the same calibration time, the fluctuation of k value and MAPE value obtained from the data with a sampling interval of five minutes is the smallest, followed by the case with a sampling interval of ten minutes. The most volatile is when the sampling interval is 15 min. In terms of calibration accuracy, with the increase in calibration time, the calibration results of three datasets with different time intervals are basically the same. It can be found from Figure 14 that the earlier the fault calibration is carried out, the greater the relationship between the accuracy after calibration and the sampling interval, and the longer the calibration time, the smaller the relationship between the two.

## 6. Comparison with Least Square Method

The least squares method is often used to calibrate drift fault [35,36]. It finds the best fitting function of the input data set by minimizing the square of error and obtains the drift slope k.

In order to demonstrate the effectiveness of the dynamic calibration method based on Bayesian inference, this method is compared with the commonly used least squares method. Figure 15 shows the calibration results under the two calibration methods. Table 3 shows the calibration slope under the two methods and the MAPE between the calibrated data and the normal value.

As can be seen from Figure 15, the difference between the calibrated data of the least squares method represented by the green line and the normal value is greater than that of the dynamic calibration method represented by the blue line. It shows that the data calibrated by the Bayesian-based dynamic calibration method is closer to the normal value. In Table 3, under the three drift amplitudes, the MAPE values obtained by the dynamic calibration method based on Bayesian inference are all smaller than those obtained by the least squares method. It shows that the dynamic calibration method based on Bayesian inference has better calibration performance than the least square method.

## 7. Conclusions

In this study, a dynamic sensor drift fault calibration method based on Bayesian inference is proposed for HVAC systems. The method has two steps. The first step is to detect the fault with a high-precision algorithm, shorten the lag time and improve the calibration accuracy. In the second step, the drift fault is calibrated by the dynamic calibration method based on Bayesian inference to complete the fault calibration from the field perspective. The proposed method is verified by using the fault dataset of a chilled water supply temperature sensor of simulation and actual chiller. Main conclusions are as follows.

(1)Through the combination of the dynamic calibration method based on Bayesian inference and the fault detection method of the EWMA control chart, the detection accuracy method is better than the other two methods—Laida criterion and Box-plot—so as to shorten the lag time and improve the calibration accuracy.(2)After the dynamic calibration method based on Bayesian inference is used to calibrate various drift faults, the MAPE value between the calibrated data and the normal data under the actual data is less than 5%.(3)When the dynamic method based on Bayesian inference is used to calibrate the drift fault, the shorter the calibration time, the lower the calibration accuracy, and the greater the influence of the sensor sampling interval on the calibration accuracy.

## Figures and Tables

**Figure 1 sensors-22-05348-f001:**
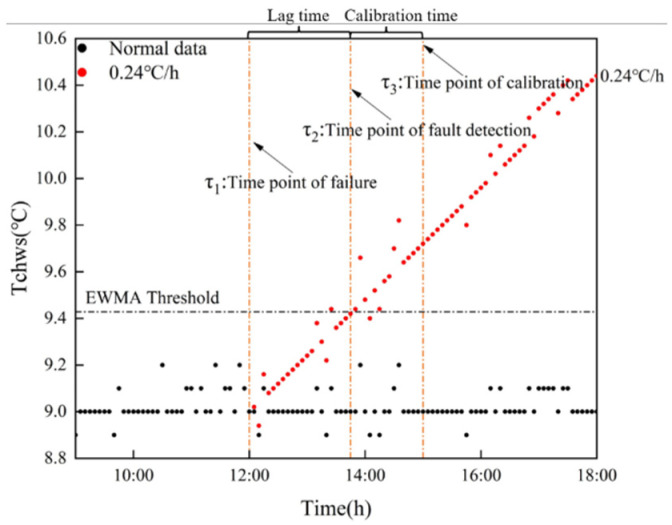
Lag time and calibration time.

**Figure 2 sensors-22-05348-f002:**
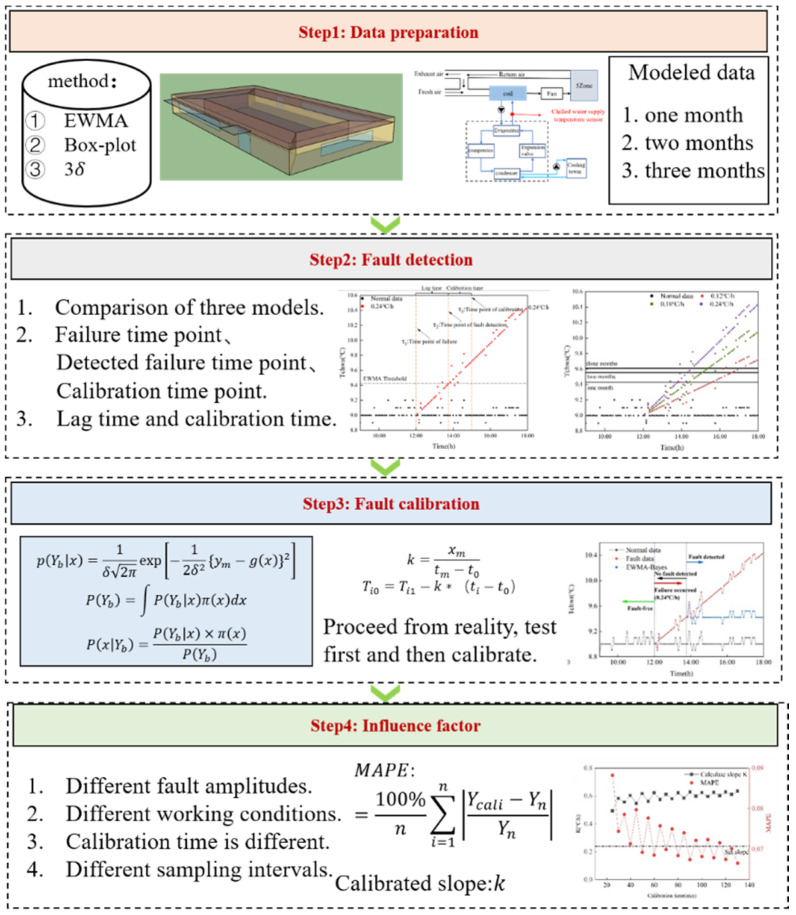
Research framework.

**Figure 3 sensors-22-05348-f003:**
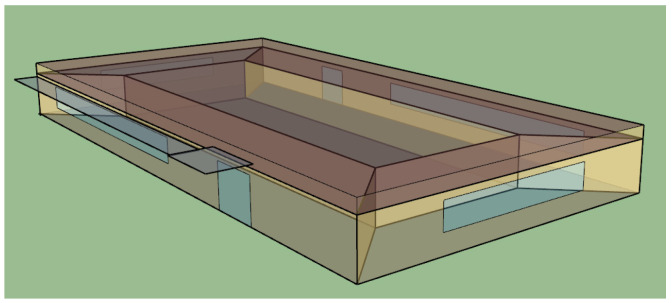
Target building.

**Figure 4 sensors-22-05348-f004:**
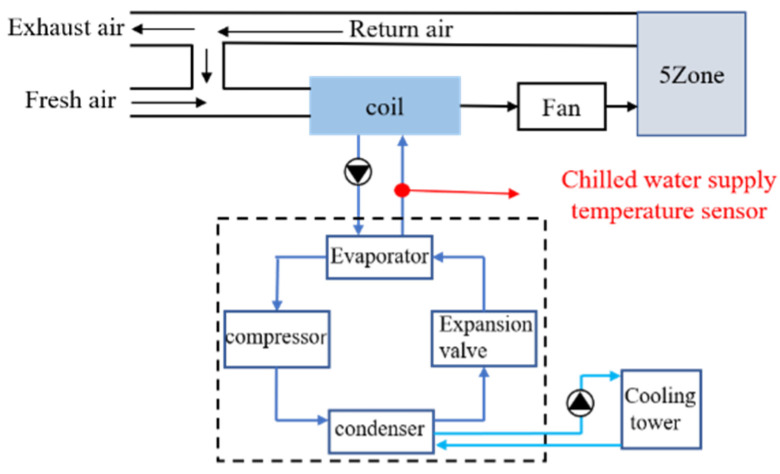
System diagram of chiller.

**Figure 5 sensors-22-05348-f005:**
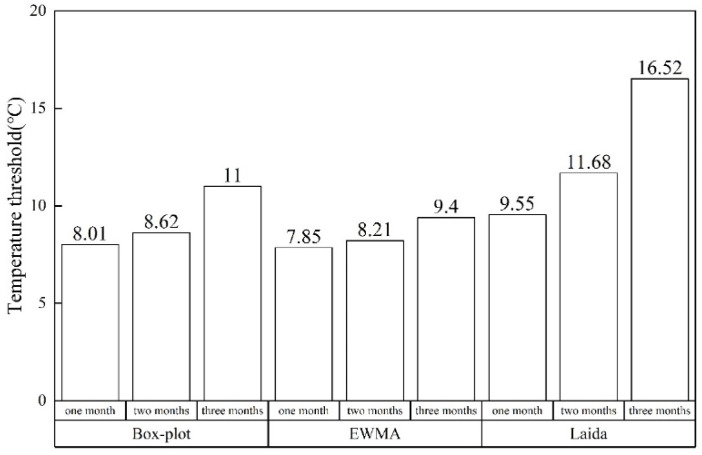
Detection thresholds of three detection algorithms.

**Figure 6 sensors-22-05348-f006:**
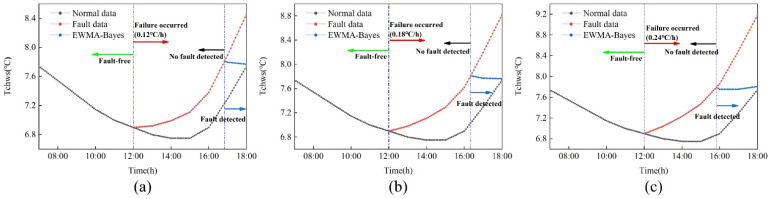
Calibration results under three drift slopes of simulation data. (**a**) 0.12℃/h; (**b**) 0.18℃/h; (**c**) 0.24℃/h.

**Figure 7 sensors-22-05348-f007:**
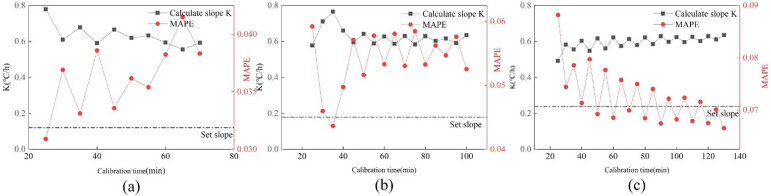
Changes of K value and MAPE value under three drift faults. (**a**) 0.12 °C/h; (**b**) 0.18 °C/h; (**c**) 0.24 °C/h.

**Figure 8 sensors-22-05348-f008:**
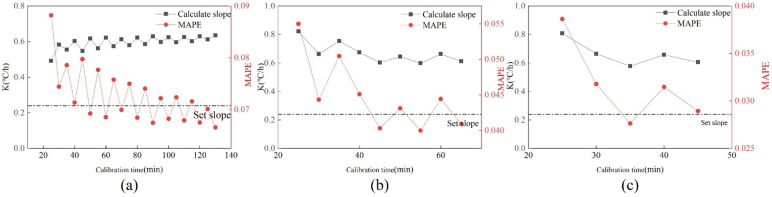
Changes of *k* value and MAPE value at three different time intervals. (**a**) 5 min; (**b**) 10 min; (**c**) 15 min.

**Figure 9 sensors-22-05348-f009:**
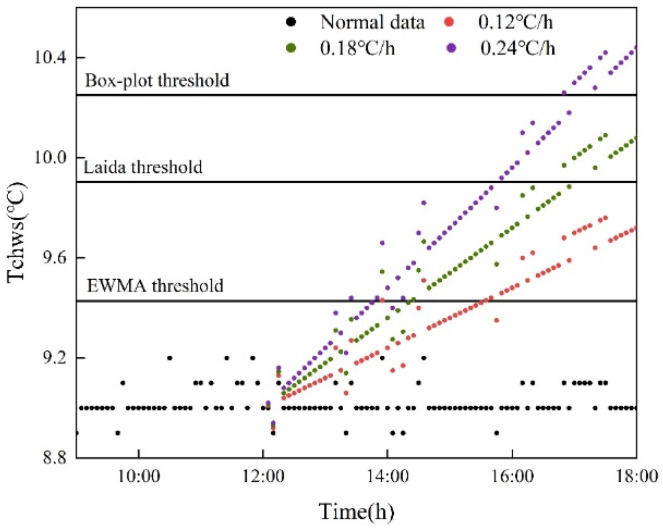
Test results under three drift slopes.

**Figure 10 sensors-22-05348-f010:**
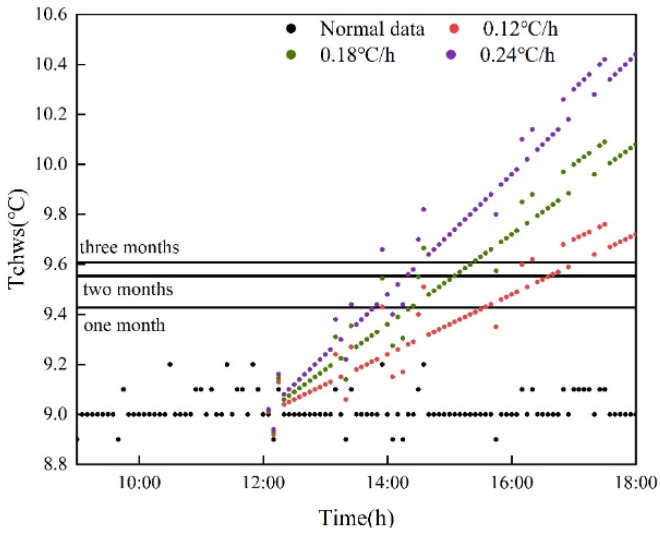
Test results of three different modeling data quantities.

**Figure 11 sensors-22-05348-f011:**
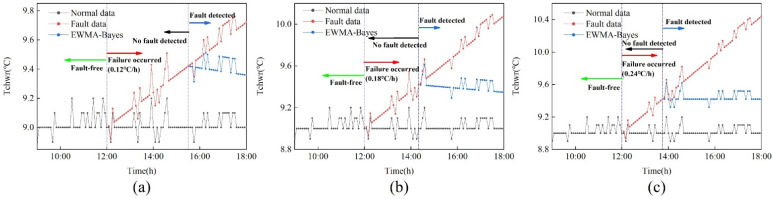
Calibration results under three drift slopes of actual data. (**a**) 0.12 °C/h; (**b**) 0.18 °C/h; (**c**) 0.24 °C/h.

**Figure 12 sensors-22-05348-f012:**
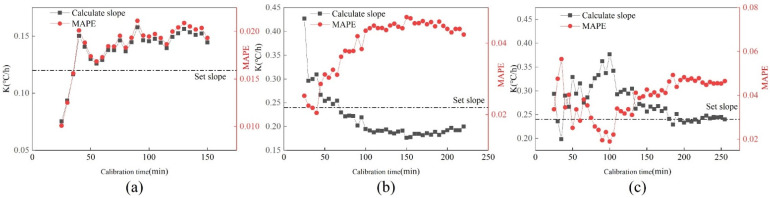
Changes in K value and MAPE value of three drift faults under the cooling tower. (**a**) 0.12 °C/h; (**b**) 0.18 °C/h; (**c**) 0.24 °C/h.

**Figure 13 sensors-22-05348-f013:**
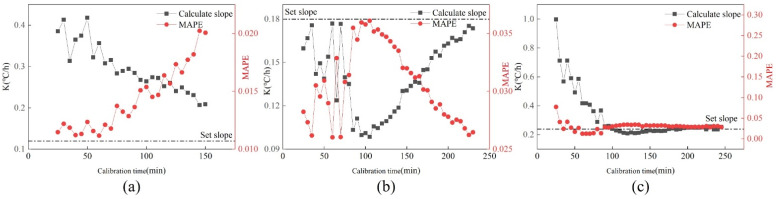
Changes in K value and MAPE value of three drift faults under a subway. (**a**) 0.12 °C/h; (**b**) 0.18 °C/h; (**c**) 0.24 °C/h.

**Figure 14 sensors-22-05348-f014:**
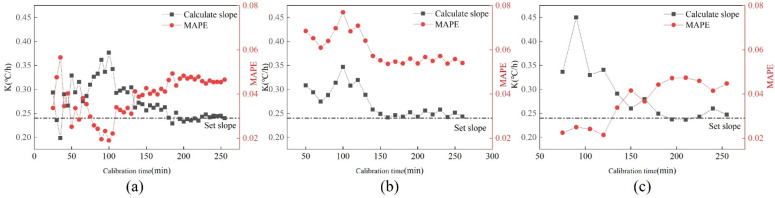
Changes in K value and MAPE value at three different time intervals. (**a**) 5 min; (**b**) 10 min; (**c**) 15 min.

**Figure 15 sensors-22-05348-f015:**
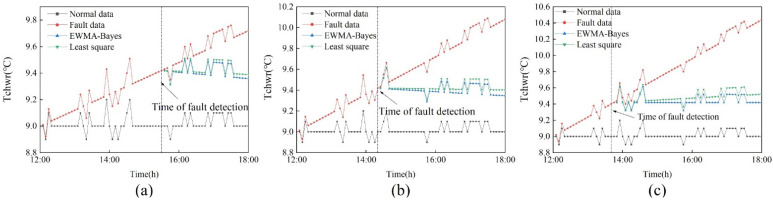
Calibration results under two calibration methods. (**a**) 0.12 °C/h; (**b**) 0.18 °C/h; (**c**) 0.24 °C/h.

**Table 1 sensors-22-05348-t001:** Calibration evaluation index of simulation data.

Set Slope	Calculate Slope *k*	MAPE
0.12 °C/h	0.593	3.8%
0.18 °C/h	0.635	5.3%
0.24 °C/h	0.635	6.66

**Table 2 sensors-22-05348-t002:** Calibration evaluation index.

Set Slope	Calculated Slope *k*	MAPE
0.12 °C/h	0.144	1.93%
0.18 °C/h	0.199	4.2%
0.24 °C/h	0.24	4.6%

**Table 3 sensors-22-05348-t003:** MAPE under two calibration methods.

Set Slope	EWMA-Bayes	Least Squares
0.12 °C/h	1.93%	4.2%
0.18 °C/h	4.2%	4.56%
0.24 °C/h	4.6%	5.87%

## Data Availability

Not applicable.

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
