# Peer review of "Dynamic Calibration Method of Sensor Drift Fault in HVAC System Based on Bayesian Inference"

_sensors, 2022, doi:10.3390/s22145348_

Round 1

Reviewer 1 Report

This paper is not suitable enough to be published in its current form, this journal template must be used. However, I would suggest that the paper should be Minor revised due to some of the corrections I pointed out in the attached and in order to raise the standard of this paper. The English needs to be polished, punctuation mark needs to be administered after each equation. Finally, I will be available for further revision of this paper.

Reviewer 2 Report

This paper proposed the study on the improvement of the traditional calibration method based on Bayesian inference to realize the dynamic calibration of sensor drift fault.

1. Please briefely review some related literatures on the drift fault calibration in the HVAC fields.

2. Plesae briefely introduce how the simulated chiller system was established.

3. Why chose the drift slopes of 0.12 ℃ / h, 0.18 ℃ / h and 0.24 ℃ / h? Please explain.

4. The fonts are extremely small and the picture is not very clear in Fig.5,Fig.10,Fig.11,Fig.12, Fig.13.

Reviewer 3 Report

This paper proposed a dynamic calibration method of sensor drift fault in HVAC system based on Bayesian inference. The results showed that the improved dynamic calibration method can effectively calibrate the drift fault of the water supply temperature sensor. However, there are some serious issues in this paper, which must be solved before possible acceptance of this paper.

1. The novelty of the paper is not clear as all the methods and algorithms are based on existing methods instead of proposed new algorithm. Therefore, there is no novel contribution in the method of this paper.

2. There is no comparison results with existing up-to-date methods to show the effectivness of the approach.

3. The literature review is not enough to support the current motivation of this paper. As claimed by author "at present, there is little research on drift fault calibration", however, little research doesn't mean no research. Therefore the problem investigated in this paper is not novel.

4. The quality of figures are not acceptable due to low resolution. Some figures even contain Chinese characters.

5. The writing of the paper must be improved as there are too many grammar issues in this paper, such as "esearch at home and abroad".

Reviewer 4 Report

The figures are hard to read

Author Response

Response: Thanks for the comment. According to the opinions, we have modified almost all figures according to the figures s format in the papers published in the sensors journal. By doing so, we hope that the revised figure can meet the high quality requirements of sensors journals. The revised part can be seen in the revised manuscript.

The revised version is sent to you as an attachment

Round 2

Reviewer 2 Report

No additional comment.

Reviewer 3 Report

All my suggestions/questions are answered by authors and this paper can be accepted in current form.